# Study on the Ultimate Load Failure Mechanism and Structural Optimization Design of Insulators

**DOI:** 10.3390/ma17020351

**Published:** 2024-01-10

**Authors:** Yongchao Ji, Zhuo Li, Peng Cao, Xinyu Li, Haoyu Wang, Xiaorui Jiang, Limin Tian, Tao Zhang, Hao Jiang

**Affiliations:** 1College of Science, Inner Mongolia University of Technology, Hohhot 010051, China; 20201000003@imut.edu.cn (Y.J.); lizuo@imut.edu.cn (Z.L.); 20231100085@imut.edu.cn (L.T.); 20231100087@imut.edu.cn (T.Z.); 20221000015@imut.edu.cn (H.J.); 2Faculty of Architecture, Civil and Transportation Engineering, Beijing University of Technology, Beijing 100124, China; caopeng@bjut.edu.cn; 3School of Mathematics and Physics, University of Science and Technology, Beijing 100084, China; bh201006@163.com; 4School of Civil Engineering, Hebei University of Engineering, Handan 056000, China

**Keywords:** disc suspension porcelain insulator, bending strength, structural optimization design

## Abstract

This study aims to enhance the productivity of high-voltage transmission line insulators and their operational safety by investigating their failure mechanisms under ultimate load conditions. Destructive tests were conducted on a specific type of insulator under ultimate load conditions. A high-speed camera was used to document the insulator’s failure process and collect strain data from designated points. A simulation model of the insulator was established to predict the effects of ultimate loads. The simulation results identified a maximum first principal stress of 94.549 MPa in the porcelain shell, with stress distribution characteristics resembling a cantilever beam subjected to bending. This implied that the insulator failure occurred when the stress reached the bending strength of the porcelain shell. To validate the simulation’s accuracy, bending and tensile strength tests were conducted on the ceramic materials constituting the insulator. The bending strength of the porcelain shell was 100.52 MPa, showing a 5.6% variation from the simulation results, which indicated the reliability of the simulation model. Finally, optimization designs on the design parameters P1 and P2 of the insulator were conducted. The results indicated that setting P1 to 8° and P2 to 90.062 mm decreased the first principal stress of the porcelain shell by 47.6% and Von Mises stress by 31.6% under ultimate load conditions, significantly enhancing the load-bearing capacity. This research contributed to improving the production yield and safety performance of insulators.

## 1. Introduction

Insulators are crucial insulating components in high-voltage transmission lines. Based on material, insulators are categorized into three main types: electrical porcelain, glass, and composite insulators [1,2]. Pollution-resistant disc suspension porcelain insulators share their primary functions with other types: firstly, ensuring electrical insulation between high-voltage transmission lines and pylons; secondly, providing mechanical fixation for the transmission lines to the pylons [3,4]. Regardless of the type, insulators must meet various electrical and mechanical performance requirements. For instance, under specified operating voltages, lightning overvoltages, and internal overvoltages, an insulator should not experience a breakdown or surface flashover. Similarly, under specified long- and short-term mechanical loads, they should not suffer damage or destruction [5,6,7,8,9]. Damaged insulators in operation can cause failures in the entire transmission line, affecting the normal functioning of the power system and posing serious threats to the safety of residents and property around the transmission lines. However, effective electrical insulation is achieved with insulators with relatively complex structures. Similarly, for good mechanical performance, insulators must have significant strength and durability to withstand adequate dynamic forces in operational conditions. Since complex structures inevitably cause stress concentrations, conducting structural analysis of the pollution-resistant disk suspension porcelain insulators and design optimization are essential to enhancing their safety performance [10,11,12,13,14,15,16,17].

The analysis of insulator electrical insulation performance and the distribution of surrounding electrical and magnetic fields has long been a focal point in insulator-related studies [18,19]. However, detailed investigation into the mechanical properties of insulators is relatively sparse. Ehsani et al. reported a comprehensive study on the mechanical, thermal, dynamic, and electrical properties of insulator materials [20]. Pilan et al. validated the effectiveness of insulators under actual operating conditions using numerical simulations and experimental tests [21]. Scholars have also made significant contributions to non-destructive testing (NDT) of insulators [22]. For example, Kim et al. introduced the Frequency Response Function (FRF) as a unique non-destructive analysis method for frequency analysis of insulators. Coupled with 3D computed tomography (3D-CT) for fault analysis, the method detected the insulator’s voids and cracks. In published reports, various non-destructive techniques (NDT) were employed for diagnosing defects in insulator components [9]. Liu et al. proposed an improved model based on YOLO for detecting insulator faults in aerial images against complex backgrounds [23]. Some researchers have also studied the static and dynamic mechanical properties of insulators. For example, Han utilized ANSYS/NASTRAN software to simulate the mechanical stresses at the interface between porcelain insulators and cement expansion in overhead transmission lines and concluded that the volume expansion of cement under load had a significant influence on the insulators’ mechanical failure [24]. De Tourreil investigated the mechanical performance of insulators under various loading conditions, including static tension, dynamic bending, pulse tension, and a combination of dynamic bending and static tension. His findings emphasized the substantial impact of end-fitting design on the insulators’ response to dynamic loads [25]. Epackachi et al. conducted a series of experiments to study the static and dynamic mechanical behavior of insulators and developed a computational model. In addition to impact hammer tests, tensile and cyclic quasi-static tests were conducted to assess the mechanical performance of insulators under transverse forces at different stages of damage. The results of impact hammer tests were used to calculate the modal frequency and corresponding viscous damping ratios for both undamaged and damaged post-insulators. Based on the mechanical behavior, an analytical model was developed to simulate the response of undamaged and damaged column insulators and to validate the experimental findings [26].

Previous studies have extensively examined the properties of insulators from various perspectives, including electrical and dynamic characteristics [27,28,29,30,31,32,33]. However, a notable research gap exists regarding the investigation of stress distribution within insulator components under ultimate loading conditions. Additionally, limited studies have focused on the mechanical aspects of insulators, which is crucial for rational optimized design analyses. By exploring the static mechanics and analyzing the stress distribution of the insulator’s components under ultimate tensile loading, a quantitative analysis can be performed through stress–strain contour maps, and subsequently, the insulator’s failure mechanisms can be quantified. These efforts can pave the way for rational optimization designs, significantly improving the qualification rate of insulator production and ensuring the reliability of normal service.

This study investigated the insulators’ behavior subjected to ultimate loading conditions. First, destructive tests were conducted. The entire destruction process was recorded using high-speed photography, and the strain was collected from specific parts of the insulator, facilitating comparative data for subsequent simulation calculations. Subsequently, static simulation analysis was performed to yield stress cloud maps of various components that accurately identified stress concentration regions. Finally, the simulations provided precise ultimate stress values. This analysis offered insights into the failure mechanisms of insulators under ultimate loading conditions. To validate the reliability of the simulation calculations, experimental tests were conducted on the materials comprising different insulator components. The reliability of simulation calculations was verified by comparing experimental results with simulation outcomes. Finally, using the simulation platform, the insulator’s structural optimization was conducted to derive the optimal design parameters for insulators.

## 2. Insulator Tensile Testing

The insulators were subjected to a tensile test using a 100-ton tensile testing machine(Jinan Tianchen Experimental Machine Manufacturing Co., Ltd., Jinan, China) at a pulling speed of 0.5 mm/min until failure occurred. To capture the strain signals during tensile testing, strain gauges were mounted to the ball pin, socket cap, and porcelain shell. Additionally, the insulators’ failure process was dynamically recorded using a high-speed camera. The experimental process is shown in Figure 1. In the experiment, the model of insulator used for experimental test is XSP-550 (Inner Mongolia Jingcheng High Voltage Insulator Co., Ltd., Inner Mongolia, China). In order to facilitate the clamping of the experimental object, we designed and manufactured the clamping link made of low carbon steel (Elastic Modulus > 235 GPa). The frequency of the high-speed camera is set to 2000 Hz.

The tensile testing determined the insulator’s ultimate load, leading to its failure, and also collected strain signals during the insulator’s failure process. This strain data were used in subsequent simulation calculations.

### 2.1. Insulator Tensile Test Results

Three specimens were subjected to tensile tests, resulting in fracture loads of 551.3 KN, 575 KN, and 583 KN, respectively. The strain and loading time curves during the entire tensile testing are illustrated in Section 4.1. Images of the insulator’s failure captured by the high-speed camera are shown in Figure 2.

The on-site observations during tensile testing and recording data through a high-speed camera helped to identify a critical tensile force of approximately 400 KN, causing brittleness and partially detaching cement at the bottom of the insulator adhesive (In). This phenomenon occurred due to an adhesive bond failure between the ball pin and the cement, leading to localized damage. However, despite the localized damage, the overall structural integrity of the insulator remained unaffected. Since the primary function of the cement within the insulator was to secure the ball pin to the porcelain shell (situated in a confined cavity), local brittle fractures did not affect the insulator’s overall load-bearing capacity. A similar scenario can be observed in compressive testing of cement blocks, where macroscopic cracks appear at a specific pressure threshold. However, the structure retains a significant load-bearing capacity without immediate collapse. Once the localized detachment of the adhesive (In) in the insulator was completed and the tensile force exceeded 500 KN, a sudden failure occurred with an abrupt bursting of the porcelain shell, clearly captured by the high-speed camera images. Moreover, the images revealed a substantial amount of cement debris during the insulator’s failure, indicating a certain degree of damage to the porcelain shell and the cement at the instant of failure. The porcelain shell experienced a structural collapse, while the cement suffered localized fracture damage.

It can be inferred that during the tensile loading, the initial failure of the adhesive interface occurs between the cement and the ball pin. This results in relative slippage and localized brittle fracture, causing the detachment of the cement. As the tensile force increases, a catastrophic failure occurs within the insulator after exceeding the porcelain shell’s strength.

CT scans were conducted on specimens comprising insulators and porcelain shells to gain further insights into the insulator’s failure. The results in Figure 3 revealed minuscule internal pores within the ceramic material, with diameters of less than 3 mm. Furthermore, a statistical analysis of the porosity indicated a pore volume of 1.4% within the ceramic specimens. In contrast, larger pores were observed while examining cement specimens, with the largest being 6 mm × 3 mm. The overall porosity of the cement specimens was about 4.07%.

## 3. Numerical Simulations

It is clear from the earlier experimental findings and analysis that advancements in ceramic manufacturing and cement pouring processes are essential to enhancing the insulators’ tensile strength. This requires addressing internal pore defects to enhance the material’s load-bearing capacity. Additionally, careful consideration of the insulator’s external geometry is crucial, with a primary focus on optimizing stress distribution across its components. To tackle this challenge, this study employed a simulation-based approach to optimize the insulator’s design dimensions (P1 and P2). The study comprehensively analyzed stress and strain conditions at various locations under ultimate load conditions. By prioritizing the ultimate stress as the primary optimization objective, simulations were performed to predict the efficient forces, thereby enhancing the insulator’s tensile strength.

### 3.1. Pre-Processing

Although insulators occupy three-dimensional (3D) space, their geometric configurations, applied loads, and constraints exhibit inherent axial symmetry. This symmetry results in uniform displacements, strains, and stresses around the central axis. Consequently, insulators can be simplified into 2D models to enhance the simulations’ computational efficiency. Specialized modeling software was used to develop a 2D model of the insulator. The model was imported into the simulation software for static analysis.

The insulator comprises four distinct materials, each characterized by specific parameters, as detailed in Table 1.

The model consists of five contact pairs: (A) contact between ball pin and socket cap; (B) contact between socket cap and adhesive (Out); (C) contact between adhesive (Out) and porcelain shell; (D) contact between porcelain shell and adhesive (In); and (E) contact between adhesive (In) and ball pin. When configuring contact conditions, it is crucial to ensure adherence according to in-service conditions and address simulation convergence issues. The mesh refinement in stress concentration areas enhances the accuracy of the simulation results.

The boundary conditions were established by fixing the upper end of the ball pin. A load of 551.3 KN (determined as the insulator’s ultimate load-bearing capacity for a specific model) was applied to the lower end of the ball pin along the length direction (negative *y*-axis), as shown in Figure 4. Moreover, after multiple attempts at calculation, the global grid size was determined to be 5 mm (when the global grid size was 5 mm, the stress at the monitoring point tended to stabilize as shown in Figure 4).

### 3.2. Simulation Results and Analysis

Figure 5 shows a 3D representation of the first principal stress within the porcelain shell. The stress analysis reveals moderate stress throughout the porcelain shell, with an average of approximately 20 MPa. Notably, a distinct maximum stress concentration exists at the inner wall of the top section of the porcelain shell, reaching a peak of 94.549 MPa. In contrast, the lowest stress exists on the outer wall of the top section, with a value of −31.279 MPa.

Figure 6 illustrates the von Mises stress distribution within the porcelain shell. The figure highlights the maximum stress concentration of about 91.108 MPa along the inner wall at the apex of the porcelain shell. The stress level on the outer wall of the apex is 60.606 MPa. Additionally, a notable stress concentration is observed along the inner wall of the neck of the porcelain shell, peaking at 63.133 MPa. Stress magnitudes in other critical areas are approximately 45 MPa, while the minimum stress (0 MPa) appears at the bottom of the skirt.

Figure 7 shows the stress distribution map, specifically highlighting the first principal stress in the adhesive (In). The stress levels within the inner wall of the adhesive (In) exceed those within the outer wall by about 49.367 MPa. A significant stress concentration is apparent at the lower end of the adhesive (In), reaching a peak value of 298.32 MPa. Figure 7 shows a pronounced stress concentration on the inner wall at the adhesive’s lower end. This concentration is primarily due to the higher plasticity of the ball pin compared to that of the adhesive (In). Suppose the stress at this specific location exceeds the ultimate limit of the adhesive (In), it may result in either localized slippage between the adhesive (In) and the ball pin bonding surface or localized brittle fracture and detachment of adhesive (In), without causing a comprehensive failure of the insulator. Furthermore, the prevailing failure mode involves the brittle fracture of the porcelain shell, accompanied by a minor detachment of the adhesive (In), while the integrity of other components remains unaffected. Consequently, localized stress overload in adhesive (In) is not the primary cause of insulator failure.

The stress distribution based on the first principal stress contour in Figure 5 reveals a distinct pattern. The internal region at the apex of the porcelain shell experiences concentrated tensile stress, while the outer wall at a comparable height undergoes compressive stress. This stress distribution resembles the pattern observed in simply supported beams subjected to bending moments. Therefore, considering the stress distribution characteristics across different components of the insulator and the material properties of each component, it can be inferred that the failure of the insulator primarily occurs when the porcelain shell exceeds its bending capacity.

## 4. Simulation Validation Experiments

### 4.1. Verification of Simulated Strain

In Section 2.1, strain data were collected for insulator components during tensile testing. A comparative analysis was conducted between simulated and experimentally obtained strains, as shown in Figure 8 and summarized in Table 2. In Figure 8, the *Y*-axis denotes the longitudinal direction (tensile direction), while the *X*-axis corresponds to the transverse direction (perpendicular to the tensile direction).

Figure 8 shows distinct strain curves for the socket cap, ball pin, and porcelain shell. The socket cap and ball pin demonstrate notable linearity, attributed to the exceptional material’s plasticity and coordination of its effective deformation. In contrast, the strain curve of the porcelain shell exhibits slight fluctuations due to the inherent high stiffness and low plasticity of ceramic materials, causing increased sensitivity to applied loads. The observed fluctuations in the strain curve of the porcelain shell anticipate minor occurrences of material instability during the insulator’s tensile process, such as potential relative slippage between the ball pin and adhesive and a localized brittle fracture in the adhesive. For instance, in Figure 8c, the ball pin demonstrates longitudinal elongation and transverse contraction, consistent with the expected results. In contrast, Figure 8a shows longitudinal and slight transverse extension in the socket cap due to the convex-concave shape at the apex of the porcelain shell. During the tensile process, the socket cap tends to deform positively in the *Y*-axis direction relative to the porcelain shell, necessitating some extension in the *X*-axis direction for overall coordination. Furthermore, as shown in Figure 8b, the porcelain shell experiences longitudinal contraction and transverse extension at the specified test point. This behavior is attributed to the inclined surface present at the measurement location. A notable difference between the simulated and experimental strains can be observed in Table 2. This difference is due to methodological variations: experimental measurements involve cumulative strain data throughout the entire tensile process, inducing a cumulative strain effect. In contrast, the simulation employed a static approach, instantaneously applying force to the model without accounting for the cumulative strain effect. Overall, the observed variance in strain between the simulation and experiment remains within an acceptable margin of less than 15%.

### 4.2. Verification of Simulation Stress

Experimental objective: The experiment was conducted to determine the capacity of the porcelain shell to withstand bending forces that can reach the strength limit followed by insulator failure. This was accomplished by conducting a comprehensive simulation study. Subsequently, experiments were performed to evaluate the tensile and bending strengths of the porcelain shell, aiming to validate the accuracy of the simulated stress results.

Principle of bending strength experiment: The bending strength experiment focused on assessing the insulator’s porcelain shell material, which was shaped into designated ceramic cylindrical specimens. These specimens were affixed to the testing platform, and a controlled force was applied by a top-loaded pressure load. The objective of the experiment was to measure the stress values at the fracture. This procedure was crucial in verifying the accuracy of the simulated stress outcomes.

The bending strength σ of the material is expressed as follows:(1)$$\sigma =\frac{M}{W}$$
where *M* is the maximum moment produced by the fracture load *P*, and *W* is the flexural section modulus of the specimen.

For specimens with a circular cross-section:(2)$$M=\frac{1}{4}PL$$
(3)$$W=\frac{\pi d^{3}}{32}$$
where *P* is the load at the fracture point of the specimen (N), *L* is the support span (mm), and *d* is the radius of the circular cross-section of the specimen (mm). Therefore, for specimens with a circular cross-section, the bending strength can be formulated as:(4)$$\sigma =\frac{8PL}{\pi d^{3}}$$

The experimental principle is illustrated in Figure 8.

The experimental results indicate that the ceramic’s tensile strength is below 3 MPa, slightly lower when compared to typical ceramics (The experimental process is shown in Figure 9. In the figure, ”SA4” represents the fourth specimen of white glazed ceramics. The experimental data is shown in Table 3). This difference can be due to the non-standard dimensions of the ceramic specimens used in this study. Non-standard dimensions can induce localized stress concentration, which may result in a lower measured strength. In contrast, simulation results show a stress in ceramics exceeding 90 MPa. Despite possible simulation errors, the occurrence of such a significant disparity is unlikely. While considering the reliability of the simulation model, it can be implied that the ceramics’ tensile strength may not be a decisive factor in constraining the insulators’ strength. Regarding bending strength, the glazed ceramics exhibit an ultimate bending strength of 100.52 MPa (The experimental data is shown in Table 4). In comparison, simulation calculations for the insulator, subjected to a maximum load of 550 KN, show a peak value of the first principal stress in the porcelain shell section at 94.549 MPa, differing by 5.49%. Notably, ceramics, being inherently brittle with limited plasticity, exhibit a bending strength that is times higher than the benchmark strength [34,35,36,37,38], which results from a trade-off between the bending strength and fracture toughness [39,40]. Therefore, it can be inferred that the insulator’s failure occurs when the ceramic material exceeds its bending strength limit. The simulation results show a close correlation with the in-service conditions, demonstrating an error margin of 5.49%.

## 5. Simulation Validation Experiments

In the previous section, a thorough analysis was conducted to explore the causes of insulator failure under ultimate tensile stress. The findings in this study revealed that an insulator fractures when the applied forces cause the failure stress that exceeds the ceramic bending strength. In the following section, simulation methods are employed to study the computation of two key design variables associated with insulators, aiming to enhance the insulator’s load-bearing capacity.

### 5.1. Introduction to Design Optimization

Optimization refers to maximizing or minimizing the design objectives under a set of given constraints. Design optimization focuses on achieving a solution that meets all design requirements while minimizing associated costs. Design optimization commonly employs two primary analytical approaches. Analytical methods involve solving differentials and extreme values to identify the optimal solution. The numerical methods use computational tools and finite elements to iteratively find the optimal solution. Analytical methods are typically employed in theoretical research, while structural optimization algorithms are frequently utilized to address complex engineering challenges [41,42,43,44,45].

The explanation of the design optimization process is shown in Figure 10.

### 5.2. Optimization Process and Analysis of Results

Variable P1 is the inclination angle of the neck at the inner wall of the porcelain shell, ranging from 8 to 14 degrees, as shown in Figure 11.

Variable P2 is the pouring height of the adhesive (In), ranging from 75 to 95 mm, as shown in Figure 11.

The porcelain shell is the most susceptible component among all insulator components; this study focused on minimizing the stress on the porcelain shell. The objective function is defined by two key parameters. P3 is the principal stress of the porcelain shell, and P4 indicates the von Mises stress of the porcelain shell. Ten sets of design samples were produced, incorporating specific variables and adhering to the defined objective function, as outlined in Table 5.

Subsequently, response surfaces and goodness-of-fit tables were constructed considering the sample points. These are presented in Table 6.

The response surface fitting for sample points P3 and P4 is given in Table 6 (The "star" in the table represents good data indicators), resulting in an R-squared value of 1, which indicates an exceptional level of fitting. However, it is important to acknowledge that goodness-of-fit measures the alignment between the response surface and the sample points and may not necessarily reflect its fidelity to the actual situation. One should carefully differentiate between these two aspects.

Figure 12 shows the 3D response surface generated by the system, where the proximity to the blue-shaded region indicates higher confidence levels. Meanwhile, Figure 13 (The pink dashed line in the figure is the boundary between P3 and P4) shows that the sensitivity of the target function P3 to parameter P2 is more pronounced. Specifically, modifying the pouring height of the adhesive (In) significantly impacts the first principal stress of the porcelain shell.

Utilizing diverse evaluation criteria, the simulation system directs the generation of three optimal candidate points, as outlined in Table 7.

Candidate point 1 is the result of system optimization. The fourth row is designated as candidate point 1 (verified), representing the candidate point validated through static calculations. This validation process is equally applicable to the remaining candidate points. In Table 7 (The ”star” in the table represents good data indicators), columns P3 and P4 indicate the percentage deviation of the objective function values from the corresponding values of the selected target point.

Candidate point 1 was discarded due to notable deviations in its validation values. Regarding candidate point 3, parameters P1 and P2 suffered alternations compared to the original model and incurred substantial engineering expenses, causing their exclusion from further consideration. Candidate point 2 exhibits superior performance across diverse metrics, maintaining consistency in parameter P1 and incurring minimal engineering expenditures. Thus, candidate point 2 was selected as the ultimate optimization outline (named the preferred point).

Subsequently, the data from the preferred point were transferred to the simulation model for reconstruction and a complete rerun of the simulation computation. Upon completion, the results of the static analysis for the optimized model were obtained.

A comparison between Figure 14 and Figure 15 reveals minimal shifts in the concentrated distribution of principal stresses within the porcelain shell. However, the maximum stress decreases significantly from the original value of 94.549 MPa to 49.481 MPa, indicating a substantial reduction of 47.6%. In contrast, stress in other critical regions increases. In summary, the stress distribution within the porcelain shell becomes more uniform, resulting in a notable improvement in material utilization. Furthermore, a comparison between Figure 15 and Figure 6 reveals a clear reduction in the maximum von Mises stress within the porcelain shell, decreasing from 91.108 MPa to 61.997 MPa, indicating a reduction of 31.9%.

## 6. Conclusions

A tensile strength test conducted on a specific insulator model reveals the highest tensile force of 551 KN. The failure images indicate complete detachment of the porcelain shell, while the other components remain largely intact. These findings suggest that the porcelain shell acts as the weakest link among the insulator components.Simulation results reveal that under an ultimate load of 551 KN, various components of the insulator exhibit notable stress concentrations. Specifically, the porcelain shell experiences the maximum first principal stress of 94.549 MPa and the maximum von Mises stress of 91.108 MPa.The simulation results highlight a stress distribution in the neck of the porcelain shell closely resembling that of a beam subjected to bending moments under the ultimate load. The insulator’s failure can occur when the porcelain shell exceeds its bending capacity.A comparison between experimental and simulation results reveals a ceramic bending strength of 100.52 MPa, while the maximum simulated first principal stress in the porcelain shell is 94.549 MPa, with a negligible error of only 5.49%. This confirms that insulator failure occurs when the porcelain shell exceeds its bending capacity, but also highlights the significant agreement between simulation and practical situations.Leveraging the design optimization feature of simulation software, the optimal solution is derived from numerous design alternatives. This leads to a substantial enhancement of insulator strength, reducing the maximum stress in the porcelain shell from 94.549 MPa to 49.481 MPa, a decrease of 47.6%. Additionally, the Mises stress decreases from 91.108 MPa to 61.997 MPa, signifying a reduction of 31.9%. The optimization significantly enhanced the overall strength of the insulator.

## Figures and Tables

**Figure 1 materials-17-00351-f001:**
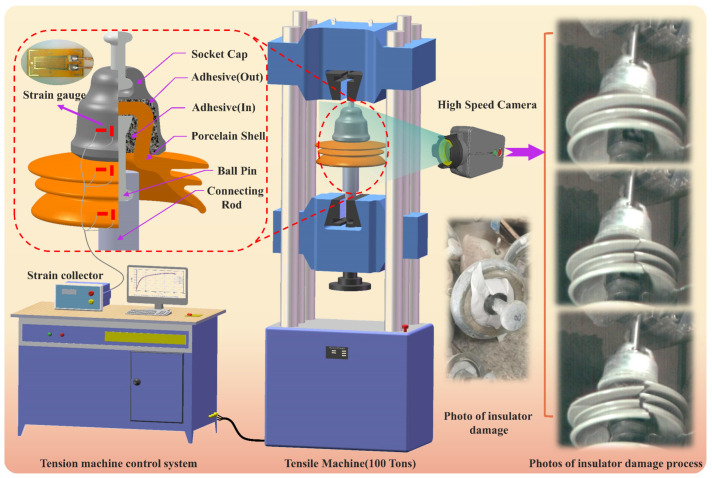
Insulator’s tensile testing.

**Figure 2 materials-17-00351-f002:**
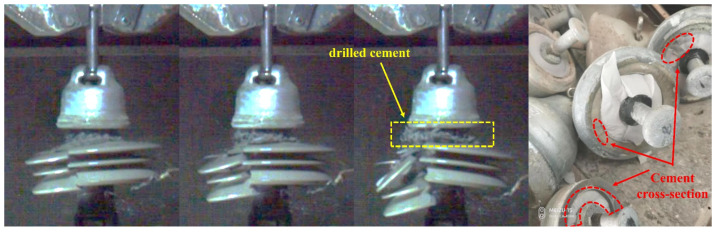
Insulator failure process.

**Figure 3 materials-17-00351-f003:**
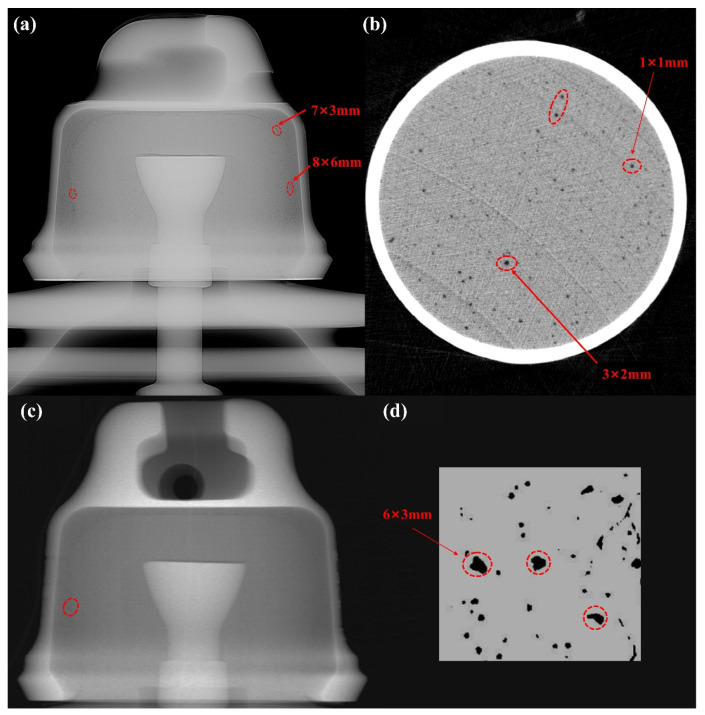
(**a**) CT scan image of insulator. (**b**) CT scan image of Porcelain shell. (**c**) Partial CT scan image of insulator. (**d**) CT scan image of Adhesive.

**Figure 4 materials-17-00351-f004:**
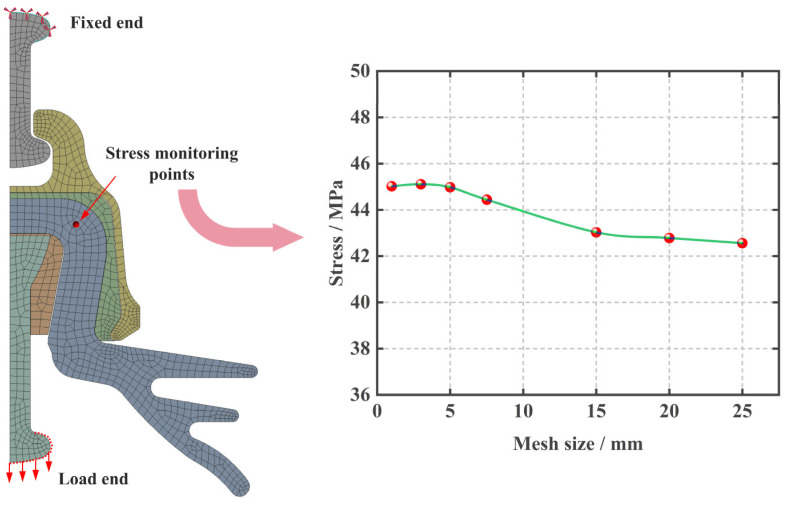
Schematic diagram for setting model boundary conditions and determining grid size.

**Figure 5 materials-17-00351-f005:**
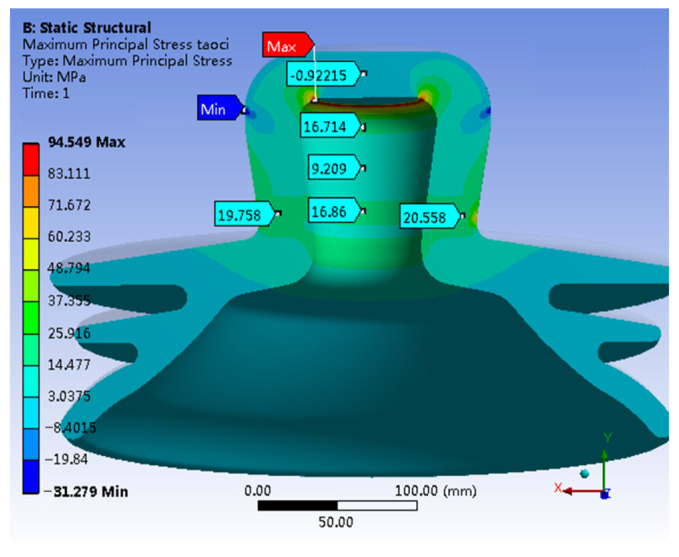
The first principal stress of Porcelain Shell.

**Figure 6 materials-17-00351-f006:**
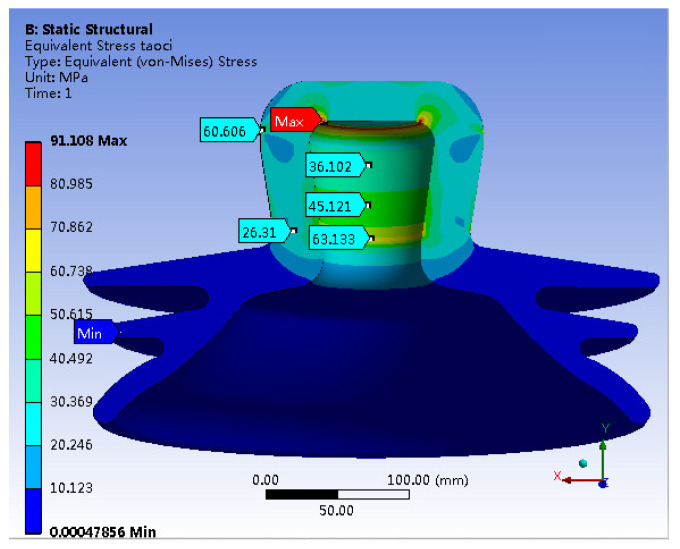
Mises Stress of Porcelain Shell.

**Figure 7 materials-17-00351-f007:**
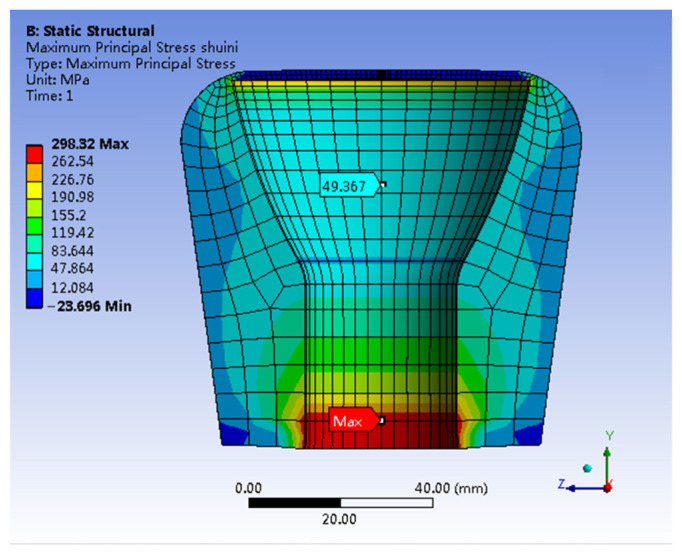
The first principal stress distribution of Adhesive (In).

**Figure 8 materials-17-00351-f008:**
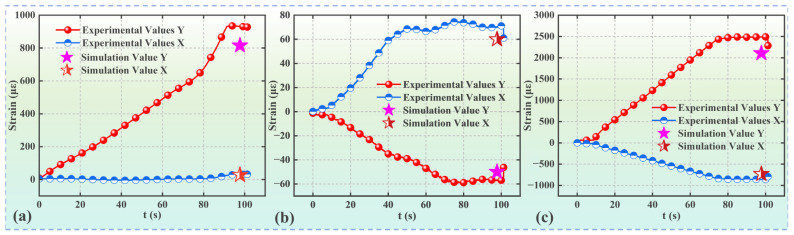
Strain data within each component of insulator: (**a**) strain data for Socket Cap, (**b**) strain data for Porcelain Shell, and (**c**) strain data for Ball Pin.

**Figure 9 materials-17-00351-f009:**
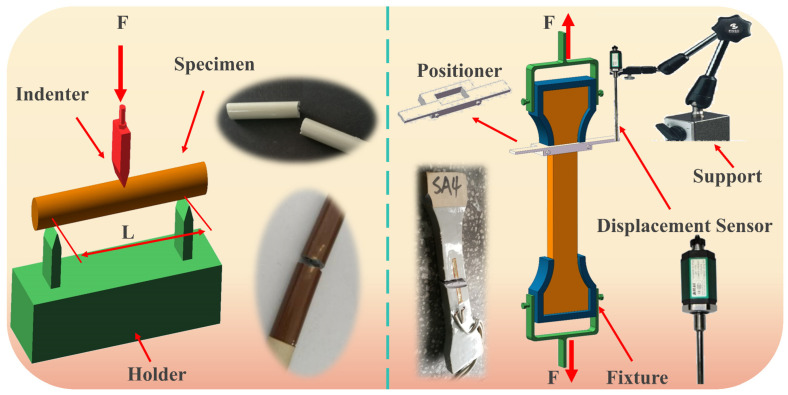
Test schematic.

**Figure 10 materials-17-00351-f010:**
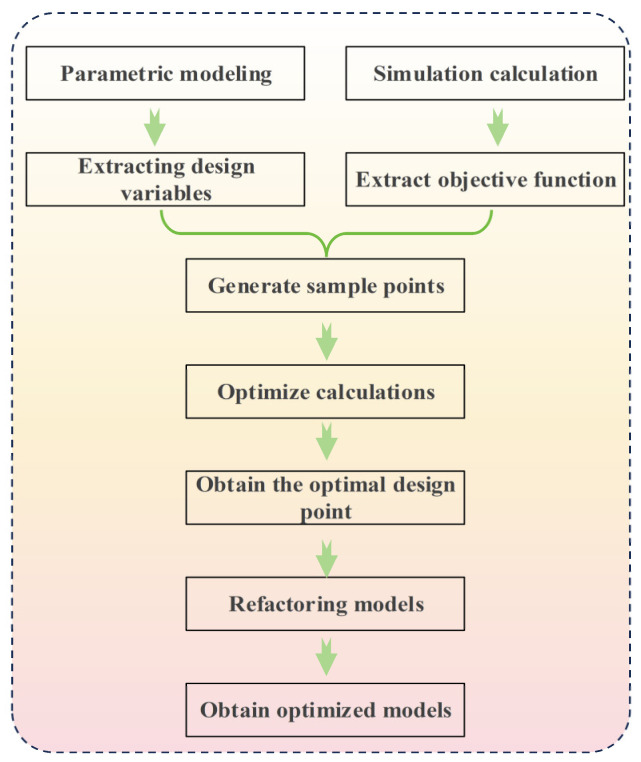
Optimize the design process.

**Figure 11 materials-17-00351-f011:**
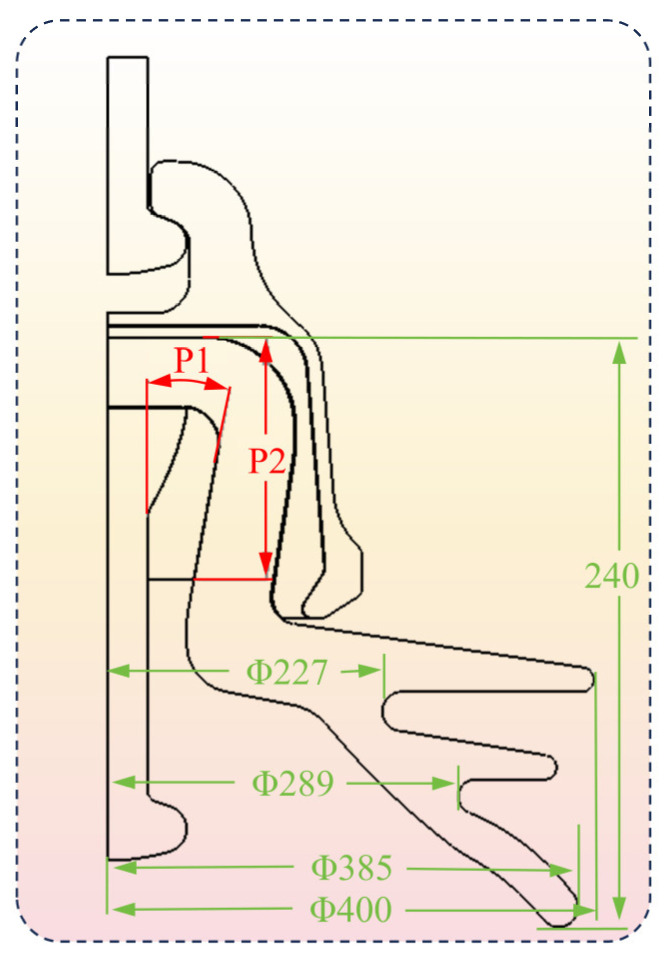
Design variable.

**Figure 12 materials-17-00351-f012:**
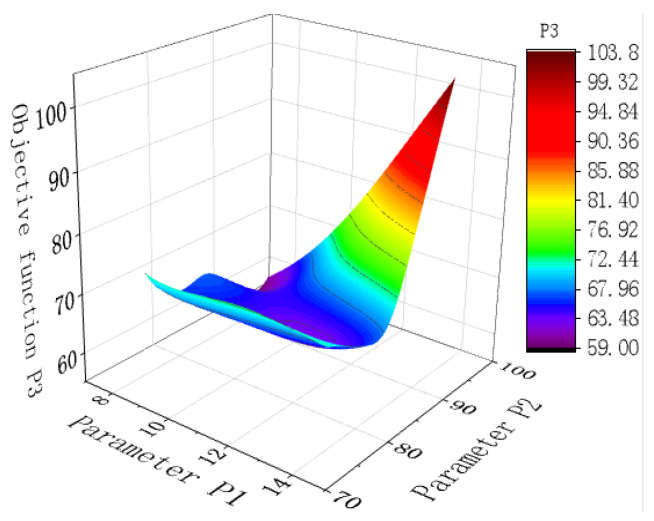
Three dimensional response surface.

**Figure 13 materials-17-00351-f013:**
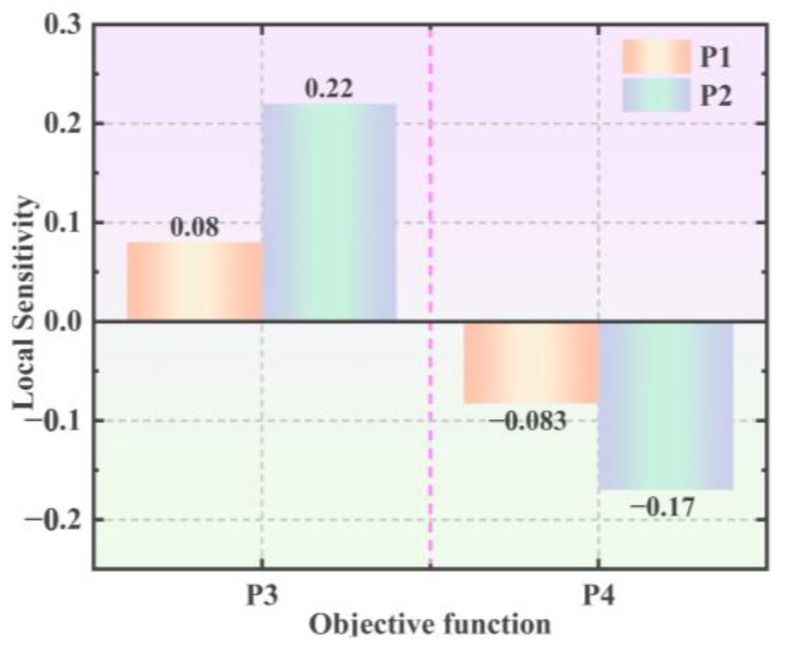
Parameter sensitivity analysis chart.

**Figure 14 materials-17-00351-f014:**
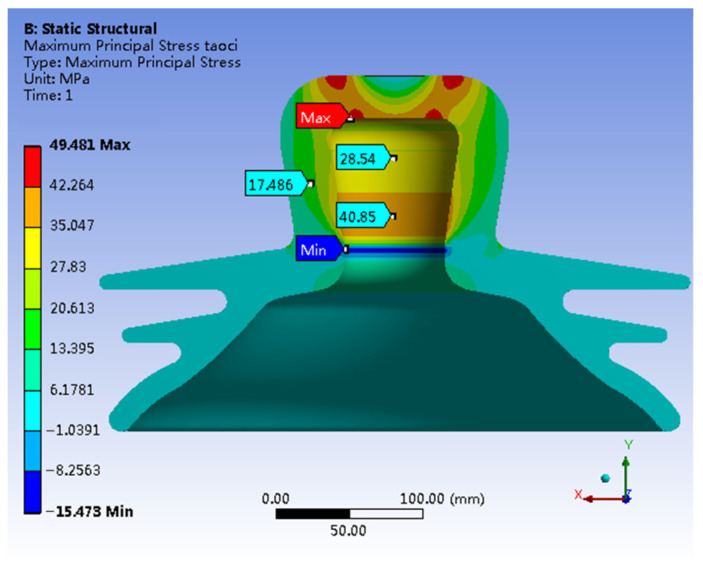
Optimizing the first principal stress of the model.

**Figure 15 materials-17-00351-f015:**
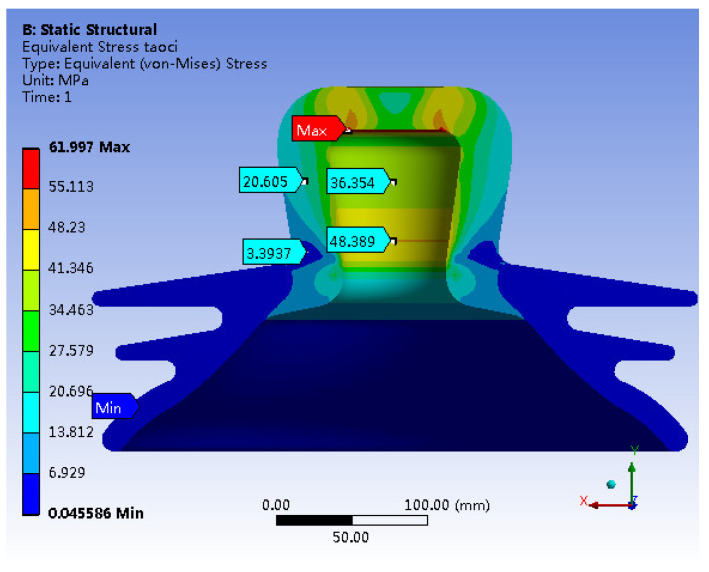
Optimizing Model Mises Stress.

**Table 1 materials-17-00351-t001:** Material parameters.

Component Name	Material	Density/(kg/m^3^)	Poisson Ratio μ	Elastic Modulus/MPa
Ball Pin	45Mn2	7800	0.269	210 × 10^3^
Socket Cap	QT450	7860	0.3	178 × 10^3^
Adhesive	Concrete	3100	0.3	40 × 10^3^
Porcelain Shell	Ceramics	3700	0.3	60 × 10^3^

**Table 2 materials-17-00351-t002:** Strain statistics.

	Socket Cap		Porcelain Shell		Ball Pin	
	Y (Max)	X (Max)	Y (Max)	X (Max)	Y (Max)	X (Max)
Experimental Values	927.8 με	32.2 με	−56.9 με	71.96 με	2490.4 με	−859.8 με
Simulation Values	846.3 με	28.3 με	−51.3 με	63.5 με	2106.3 με	−732.6 με
Error	8.7%	12.1%	9.8%	11.7%	15.4%	14.8%

**Table 3 materials-17-00351-t003:** Experimental results of tensile strength.

Sample Number	1	2	3	4	Mean Value
White glazed ceramics	2.71 MPa	2.90 MPa	3.02 MPa	3.14 MPa	2.94 MPa
Yellow glazed ceramics	2.23 MPa	2.58 MPa	2.63 MPa	3.46 MPa	2.73 MPa

**Table 4 materials-17-00351-t004:** Tensile strength results.

Sample Number	1	2	3	4	Mean Value
White glazed ceramics	105.03 MPa	104.95 MPa	92.16 MPa	99.95 MPa	100.52 MPa
Yellow glazed ceramics	94.38 MPa	94.67 MPa	95.46 MPa	104.66 MPa	97.29 MPa

**Table 5 materials-17-00351-t005:** Design sample points.

Name	Update Order	P1/°	P2/mm	P3/MPa	P4/MPa
1	4	11	85	63.224	52.091
2	2	8	85	66.921	55.455
3	6	14	85	68.696	53.173
4	3	11	75	72.981	61.666
5	7	11	95	77.969	56.952
6	1	8	75	72.576	60.827
7	5	14	75	71.715	60.021
8	8	8	95	60.547	48.078
9	9	14	95	103.78	74.333

**Table 6 materials-17-00351-t006:** Goodness-of-fit table.

1	Name	P3	P4
2	Goodness of Fit
3	Coefficient of Determination (Best Value = 1)	 1	 1
4	Maximum Relative Residual (Best Value = 0%)	 0	 0
5	Root Mean Square Error (Best Value = 0)	1.45 × 10^−7^	9.24 × 10^−8^
6	Relative Root Mean Square Error (Best Value = 0%)	 0	 0
7	Relative Maximum Absolute Error (Best Value = 0%)	 0	 0
8	Relative Average Absolute Error (Best Value = 0%)	 0	 0

**Table 7 materials-17-00351-t007:** Information table of optimal candidate points.

Name	P1	P2	P3/MPa	P4/MPa
ParameterValue	Variation from Reference	ParameterValue	Variation from Reference
Candidate Point1	8	92.6	 59.1	−1.27%	 48.3	−0.05%
Point1 (verified)	 68.7	14.82%	 57.3	18.53%
Candidate Point2	8	90.6	 59.8	0.00%	 48.3	0.00%
Point2 (verified)	 62.0	3.64%	 49.5	2.41%
Candidate Point3	8.81	88.9	 63.1	5.37%	 49.4	2.31%
Point3 (verified)	 62.4	3.88%	 49.9	3.18%

## Data Availability

Data are contained within the article.

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
