# Peer review of "Study on the Ultimate Load Failure Mechanism and Structural Optimization Design of Insulators"

_materials, 2024, doi:10.3390/ma17020351_

Round 1
Reviewer 1 Report
Comments and Suggestions for Authors
This study incorporates both experimental and numerical approaches. The feedback suggests that additional information is needed to adequately explain both the experimental and simulation aspects of the paper. It is recommended to include more details and consider revising the content before publication. A minor comment has been noted:
1. add detailed information for the experiment; specimen, jig, etc. (material, material properties, dimensions, friction, high-speed camera frequency, etc.). ※ detailed material information is presented in Table 1. Add a simple explanation in section 2.
※ dimension can be added in the Figure 10.
2. In Table 1, the material properties obtained from the test? or manufacturing mill sheet?
3. Explain FE model with illustration (element size, boundary condition, etc.). How to decide the element size? stress level is able to change depending on the element size.
4. Concerning Figure 6, there is a discrepancy in the minimum stress values, with -23.696 MPa differing from the main body's minimum stress of 0 MPa. Please investigate and address this inconsistency for clarity.
5. In Figure 7, the numerical value shows only the last value. is the simulation perturbation? it is suitable to compare with the experiment?
6. For Figure 11, it is recommended to use a transparent graph for improved visualization.
Reviewer 2 Report
Comments and Suggestions for Authors
This manuscript highlights the possibilities of enhancing the performance of high-voltage transmission line insulators and their operational safety by investigating their failure mechanisms under ultimate load conditions. The research is well-designed and presented clearly. A good comparative analysis of existing publications is carried out. The methodological section of the manuscript is presented in sufficient detail. The authors used modern equipment for the preparation and testing of samples. They also utilized the equipment for visualization and assistance in interpreting the obtained results. The authors conducted fracture tests on a specific type of insulator under ultimate load conditions. They also established a simulation model of the insulator to predict the effects of ultimate loads. By leveraging the design optimization feature of simulation software, the optimal solution was derived from numerous design alternatives. The optimization significantly enhanced the overall strength of the insulator.
However, some issues in describing the results should be addressed and some general shortcomings should be corrected to make the manuscript acceptable for publication in Materials.
(1) In the Abstract, Lines 23–24, it seems that the sentence “This implied that insulator failure resulted from the porcelain shell when it exceeded its bending limit” should be corrected. My understanding of the process is as follows: “This implied that the insulator failure occurred when the stress reached the bending strength of the porcelain shell”.
(2) The phrase in Line 180 should be deleted.
(3) Lines 265–267: The sentence “Experimental objective: The experiment was conducted to determine the capacity of the porcelain shell to withstand bending forces that exceed its limits, leading to insulator failure” should be corrected as the porcelain shell can only withstand bending forces that are less or reach the strength limit. Therefore, this sentence may be as follows: “Experimental objective: The experiment was conducted to determine the capacity of the porcelain shell to withstand bending forces that can reach the strength limit followed by insulator failure”.
(4) Line 276: The marking of the bending strength should be corrected.
(5) Line 277: M is the maximum moment produced by the fracture load P (not the maximum deflection). This issue should be fixed.
(6) In Lines 290–306, references to Table 3 and Table 4 should be made. Besides, the captions of these tables should be corrected indicating “experimental” and “simulation” results, respectively.
(7) In Lines 302–304, to substantiate the mentioned discrepancy, the corresponding sentence can be expanded as follows: “Notably, ceramics, being inherently brittle with limited plasticity, exhibit a bending strength that is times higher than the benchmark strength [34-38], which results from a trade-off between the bending strength and fracture toughness [References].” The suggested references are: https://doi.org/10.1007/BF02355530 and https://doi.org/10.1023/A:1018655917051
(8) Line 310: The phrase “when the ceramic’s bending strength exceeds its critical threshold” is confusing. Actually, the bending strength is the same critical threshold and cannot exceed itself. Therefore, the sentence may be as follows: “The findings in this study revealed that an insulator fractures when the applied forces cause the failure stress that exceeds the ceramic bending strength”.
(9) In Table 5, the measurement unit for P1 should be degree (instead of mm). This issue should be fixed.
(10) Figure 12: The vertical axis name contains a typo “Sensetivety” and should be corrected.
(11) In the caption of Figure 13, the word “Optimize” should be replaced with “Optimizing”.
Comments on the Quality of English LanguageIn my opinion, the English language of this manuscript should be slightly improved.
Round 2
Reviewer 2 Report
Comments and Suggestions for Authors
All of the reviewer’s comments were taken into account by the authors. The manuscript can now be accepted for publication in Materials.